# Characterizing mobility patterns and malaria risk factors in semi-nomadic populations of Northern Kenya

**Hannah R. Meredith**[1], **Amy Wesolowski**[2], **Dennis Okoth**[3], **Linda Maraga**[4], **George Ambani**[4], **Tabitha Chepkwony**[4], **Lucy Abel**[4], **Joseph Kipkoech**[4], **Gilchrist Lokoel**[3], **Daniel Esimit**[3], **Samuel Lokemer**[3], **James Maragia**[3], **Wendy Prudhomme O'Meara**[1,5,6☯] *, **Andrew A. Obala**[7☯]

1 Duke Global Health Institute, Durham, North Carolina, United States of America, 2 Johns Hopkins Bloomberg School of Public Health, Baltimore, Maryland, United States of America, 3 Department of Health Services and Sanitation, Lodwar, Turkana County, Kenya, 4 Academic Model Providing Access to Healthcare, Eldoret, Kenya, 5 School of Public Health, Moi University College of Health Sciences, Eldoret, Kenya, 6 School of Medicine, Duke University, Durham, North Carolina, United States of America, 7 School of Medicine, Moi University College of Health Sciences, Eldoret, Kenya

☯ These authors contributed equally to this work.
* wendy.omeara@duke.edu

**Data Availability Statement:** The minimal dataset and scripts necessary to reproduce the figures and tables in this paper are available at the Duke

## Abstract

While many studies have characterized mobility patterns and disease dynamics of settled populations, few have focused on more mobile populations. Highly mobile groups are often at higher disease risk due to their regular movement that may increase the variability of their environments, reduce their access to health care, and limit the number of intervention strategies suitable for their lifestyles. Quantifying the movements and their associated disease risks will be key to developing interventions more suitable for mobile populations. Turkana, Kenya is an ideal setting to characterize these relationships. While the vast, semi-arid county has a large mobile population (>60%) and was recently shown to have endemic malaria, the relationship between mobility and malaria risk in this region has not yet been defined. Here, we worked with 250 semi-nomadic households from four communities in Central Turkana to 1) characterize mobility patterns of travelers and 2) test the hypothesis that semi-nomadic individuals are at greater risk of malaria exposure when migrating with their herds than when staying at their semi-permanent settlements. Participants provided medical and travel histories, demographics, and a dried blood spot for malaria testing before and after the travel period. Further, a subset of travelers was given GPS loggers to document their routes. Four travel patterns emerged from the logger data, Long Term, Transient, Day trip, and Static, with only Long Term and Transient trips being associated with malaria cases detected in individuals who carried GPS devices. After completing their trips, travelers had a higher prevalence of malaria than those who remained at the household (9.2% vs 4.4%), regardless of gender and age. These findings highlight the need to develop intervention strategies amenable to mobile lifestyles that can ultimately help prevent the transmission of malaria.

Research Data Repository (DOI: 10.7924/
r49311n5m, https://doi.org/10.7924/r49311n5m).

**Funding:** This work was supported by the Fogarty
International Center (VECD Consortium Fogarty
Global Health Fellowship to HRM), the Burroughs
Wellcome Fund (Career Award at the Scientific
Interface to AW), the National Institute of Health
(Director's New Innovator Award, grant number
DP2LM013102-0, to AW), and the National
Institute of Allergy and Infectious Diseases (grant
number 1R01A1160780-01 to AW). The funders
had no role in study design, data collection and
analysis, decision to publish, or preparation of the
manuscript.

**Competing interests:** The authors have declared
that no competing interests exist.

## Introduction

Quantifying the relationship between human mobility and disease transmission is critical for
developing more effective interventions [1–5]. Most studies have focused on individuals that
travel to/from a permanent residence or mobility and disease transmission patterns that have
been generalized to larger geographic areas [1–9]. Fewer studies have focused on more mobile
individuals, such as semi-nomads, who are difficult to reach and do not follow the general
mobility patterns of the larger population. Relative to settled individuals, they are often at
higher disease risk due to their regular movement, reduced access to health care, and lack of
interventions suitable for their lifestyles [10–14]. In some settings, individuals who move regu-
larly for their livelihoods are exposed to infectious diseases more frequently than their settled
counterparts [15]. For example, nomadic pastoralists seeking water for their animals may be
around mosquito breeding sites more often, thus increasing their risk of malaria infection
[11,16]. In other settings, mobile populations' isolation and frequent movement may result in
irregular exposure to diseases circulating in the settled community [17]. While their frequent
movements could reduce initial exposure, it could also render them more susceptible to out-
breaks and worse symptoms later on due to reduced acquired immunity and low vaccination
coverage [11,17,18]. Further, eradication of infectious diseases may be challenging if transmis-
sion is concentrated in these hard-to-reach and under-served populations. For instance, both
smallpox and polio were reintroduced into settled communities by nomadic populations who
were unvaccinated [14,19]. Therefore, a better understanding of mobile populations travel pat-
terns and their relationship with disease dynamics would help determine when, where, and
who to focus on in intervention strategies and elimination campaigns.

Typically, human movements have been estimated by census data, traffic and travel surveys,
flight statistics, night-time satellite images, call data records (CDRs), social media, and per-
sonal global positioning systems (GPS) [6,20–24]. These methods have been used to study pop-
ulations that are easy to locate, own cell-phones, and use established travel networks.
However, the resulting datasets may not be relevant for characterizing the movements of
mobile, remote populations that are either difficult to reach or intentionally excluded [15,25].
Additionally, regular movements motivated by pastoralism and hunting and gathering are not
typically captured by general surveys, like censuses taken every 5–10 years, and would likely be
aggregated into larger flows of movement between administrative units (i.e., towns, districts,
regions) during mobile phone or social media data pre-processing. Thus, specific studies are
needed to characterize mobility patterns of uniquely mobile populations. For example, GPS
loggers have been used to characterize travel patterns of mobile populations in Lao PDR, Mon-
golia, and Senegal [26–28]. While a large proportion of the world's mobile populations reside
in Africa and some studies have documented the health challenges and general travel patterns
of different mobile populations across Africa [10,12,13,16], few studies have quantified the
movement patterns and possible relationship with infectious disease transmission.

Turkana is a semi-arid county in north-western Kenya with a sparse population that is 60%
semi-nomadic (where at least one household member seasonally migrates with their herd) or
nomadic [29] (**Fig 1A**). The mobile lifestyles of the Turkana have been studied from anthropo-
logical and ecological perspectives [30–32]; however, their travel patterns have not been well
quantified, largely relying on individuals recounting trip details in surveys or indicating their
routes on maps [29]. Similarly, the disease dynamics of the mobile Turkana have not been well
studied, relying on a few studies that use self-reporting of health complaints and symptoms
that could be associated with certain diseases [33,34]. While these studies attributed the Turka-
na's health complaints (or lack thereof) to their mobile lifestyles, there has yet to be a study
directly relating their travel patterns and risk of disease. To define this relationship, we

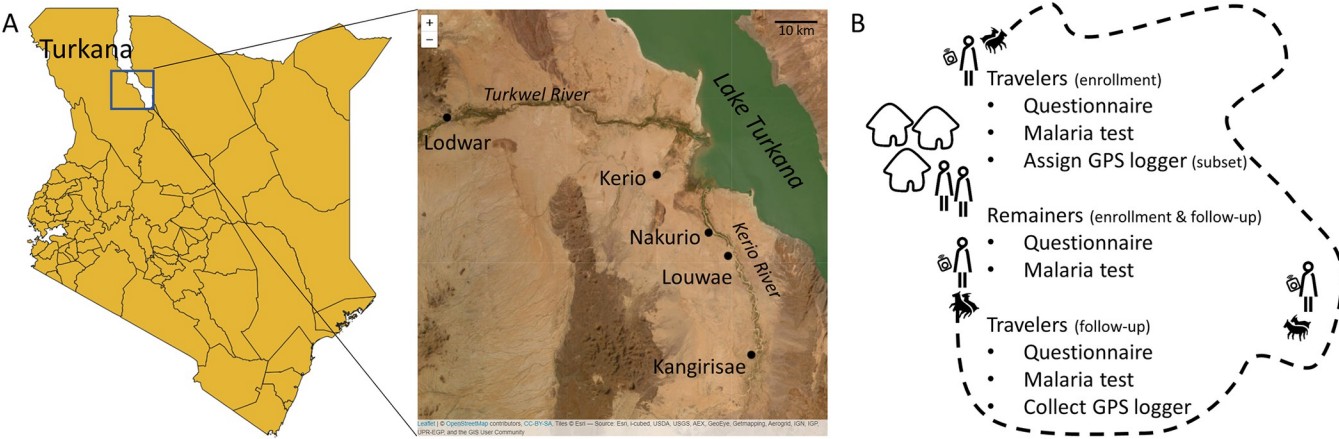

**Fig 1. Overview of study area and design.** (A) Enrollment took place in Central Turkana (box on left map), near four health facilities (labeled on right map). (B) Semi-nomadic households with at least one traveler and remainer were enrolled. Before and after travelers migrated with their herds, all consented members provided blood samples for malaria tests and answered questions on recent travel and medical history. GPS loggers were assigned to a subset of travelers. Shapefiles were downloaded from DIVA-GIS (https://www.diva-gis.org/) and Esri World Imagery was accessed via the R package leaflet.

conducted a study of semi-nomadic households across Central Turkana to better understand their mobility patterns and determine if traveling affects their risk for disease exposure. Specifically, we focused on the risk of malaria exposure because it was recently confirmed to be endemic in Central Turkana [35]. However, as the majority of the households enrolled for this previous study were settled, malaria exposure in more mobile households from this area remains to be characterized. Since semi-nomadic pastoralists are motivated to migrate in search of food and water for their herds when conditions around the homestead get too dry, they might be exposed to potential mosquito breeding sites more often than their household members who remain behind. Thus, we hypothesized that the household members who travel with their herd are more likely to have malaria after the migration than the members who remained at the settlement. If this is true, they may be importing malaria back to their villages of origin. Ultimately, understanding the extent to which mobile populations impact malaria transmission is key for informing elimination efforts by providing insight on how to better tailor surveillance and intervention strategies for these unique populations.

## Methods

### Ethics

Written informed consent was provided by all adults and by parents or guardians for individuals under 18 years old. Individuals 12–18 years old were asked for verbal assent. This study was approved by the ethical review boards of Moi University (IREC/2020/209) and Duke University (Pro00107835).

### Inclusivity in global research

Additional information regarding the ethical, cultural, and scientific considerations specific to inclusivity in global research is included in the S1 Checklist.

### Study area

Turkana is a semi-arid county in north-western Kenya (**Fig 1A**) with a population that is ~60% (semi-)nomadic pastoralist [36]. Rainfall in Turkana is short and intense, resulting in

rapid, transient vegetation growth and water accumulation in areas that travelers will seek out for their herds. While malaria is endemic in Central Turkana [35,37], the study time-frame (March-October 2021) focused on when the rains started and most households would have at least one member traveling with their herd.

We enrolled semi-nomadic households from four catchment areas defined by health facilities in Central Turkana (Kerio, Nakurio, Louwae, and Kangirisae). From our communication with community health workers, we estimated that most households in this area are semi-nomadic (>90%). The population in these catchment areas fluctuates seasonally, but the approximate number of households was 3800 (per communications with chiefs and community health workers). These catchment areas are located near the seasonal Kerio River that empties into Lake Turkana, which has alkaline water that is typically not used for drinking water (personal communication with community health workers). Generally, semi-nomadic households build semi-permanent settlements. Temporary structures made of palm fronds and saplings are present; however, household members typically sleep outside. During dry periods, part of the household travels with livestock in search of water and pasture.

## Study population

We recruited and consented semi-nomadic households with at least one person planning to travel with the herd for at least two consecutive weeks (traveler) and at least one person planning to remain behind at the homestead (remainer). Individuals had to be at least one year old to be eligible for participation. At enrollment (before the travelers left with their herds) and again at follow-up (after the travelers returned with their herds), participants provided a finger-prick blood sample for a dried blood spot (DBS) and answered a questionnaire detailing their travel and medical history (**Fig 1B**). Participants who felt unwell at enrollment or follow-up were offered a malaria rapid diagnostic test (RDT) and referred for appropriate care at nearby health facilities. RDTs were used for care, rather than assessing research outcomes.

## GPS logger substudy

One traveler per household, either the head of household or the lead herder for the household, was asked to carry a GPS logger during their trip. The number of travelers assigned a GPS logger was limited by both the number of GPS loggers available (48) and when travelers returned so a logger could be reassigned to a new household. The GPS logger (model i-gotU GT-600) was light-weight (< 80 grams), small (46x41.5x14mm), water resistant, battery powered (750 mAh), password protected, has 64 Mb memory with the capability of storing 262,000 location points, and could be worn in multiple ways (i.e., lanyard, velcro, watchband). GPS loggers were programmed to record location, date, and time eight times a day. To conserve battery power and ensure travel patterns were recorded at different times of the day, loggers were programmed to record locations hourly during two moving four-hour windows separated by 12 hours (i.e., Day 1: 12, 1, 2, 3 am and pm; Day 2: 4, 5, 6, 7 am and pm, etc.).

## GPS logger analysis

GPS tracks that covered at least 50% of travel dates, as defined by the dates between enrollment and follow-up, were included in analysis. While the dates of enrollment and follow-up did not always correspond with the departure and return dates reported by travelers, a sensitivity analysis suggests this should not affect the results (**S1 Text**). To distinguish between short movements around a given point that could be associated with stationary grazing and longer directional movements, sequential GPS points that fell within a 500m radius were hierarchically clustered using the hclust and cutree functions from the geosphere package in R

(version 4.2.2). New coordinates based on the centroid and an ID were assigned to each new cluster. Each GPS track was analyzed by plotting the cluster IDs as a function of date and time of day (i.e., night (6pm– 6am) vs day). Night locations were categorized as long-term campsites if a week or more of consecutive nights were spent there or transient campsites if fewer than a week of consecutive nights were spent there (see **S2 Text** for sensitivity analysis).

Travel trajectories from GPS loggers were analyzed individually and then categorized into four 'trip types'. Long Term trips had the majority of nights logged at long-term camps while Transient trips had the majority of nights logged at short-term camps. Day trips had >90% of night GPS points logged at the same location recorded on the evening of their enrollment, but different GPS points logged during the day. This likely reflects scenarios in which the travelers conducted day trips with their herds and returned to the same camp each night. Static trips had >90% of all points (day and night) logged at the same location, likely representative of scenarios in which the travelers stayed within a 500-meter radius for the duration of the study or the loggers were not carried. GPS points were mapped in R on Esri.WorldImagery provider tiles using the leaflet (version 2.1.1) and leaflet.esri (1.0.0) packages.

## Molecular detection of Plasmodium falciparum

Genomic DNA was extracted from each DBS using the Chelex method. All DBS extracts were screened with genus-specific primers for Plasmodium spp. Positive samples were tested for the presence of *P. falciparum* using species specific primers. The expected product size was 206 bp which was visualized on a 2% agarose gel stained with Sybr safe [38].

## Data capture and statistical analyses

Community health workers (CHWs) were trained to identify and consent eligible households and conduct the enrollment and follow-up visits. Two CHWs per catchment area (Kerio, Nakurio, Louwae, and Kangirisae) were appointed because they are trusted community members, they speak the local language, and they are in communication with the households to know their travel plans. The research team met with the CHWs regularly to refresh their skills and review every data collection tool to minimize errors and missing data. Data collected from the surveys were entered into a REDCap database (https://www.project-redcap.org/) and analyzed in R. PCR results from the DBS were saved in Microsoft Excel and imported into R for analysis.

We compared demographic, travel and medical history between the following subpopulations: 1) travelers who did or did not carry loggers to assess generalizability of the logger data, 2) those who traveled or who remained at the settlement and 3) those who did or did not acquire a malaria infection over the travel period. Bivariate and multivariable logistic regression analyses were used to quantify correlation between infection outcome and putative risk factors (glm function, R).

## Results

### Traveler population and trip details

Between March and October, 2021, we enrolled 250 households that included at least one person who expected to remain and one who expected to travel (n = 929 participants). In total there were 304 members who reported plans to travel with the herd. At follow-up, 70 of these members reported that they had not traveled, 44 additional members reported they had traveled with the herd, and 18 members were lost to follow-up, thus 260 travelers from 249 households were included in the travel analysis (**Fig 2**). The majority of travelers were male (87.7%,

228/260) with a median age of 31 years (interquartile range (IQR) 19–40) and a median trip duration of 57.2 days (IQR 42.2–76.2) (**Table 1**). Of the 260 travelers, 64 carried a GPS logger throughout the entire study period; however, only 58 tracks were analyzed because four GPS did not return data. Two GPS loggers were lost in the field, one lost battery power before the trip started, one did not consistently collect data throughout the trip, and two lasted less than 50% of the travel period.

We compared baseline demographic characteristics between those who did and did not carry a GPS logger to determine general representativeness of travelers (**Table 1**). The main differences were the carriers tended to be slightly older (37 years (IQR 30–45) vs 30 years (IQR 19–40)) and had a higher proportion of males (93.1% (54/58) vs 86.1% (174/202)) (**Table 1**). Trip details were similar across groups, although slightly more individuals with GPS loggers reported staying at a campsite with non-household members (89.7% (57/58) vs 81.7% (165/202)) (**Table 1**). Given these similarities, we assumed the GPS logger data generally represented the spatial-temporal patterns of this semi-nomadic community's trips.

### Travel pattern analysis

At the population level, GPS data revealed that travelers from the same catchment area tended to travel in common regions. Very few "hotspots", where multiple travelers trajectories intersected [39], were observed; most of the points (79.2% of night points and 71.6% of all points) were visited by a single traveler. The few hotspots that were identified tended to be near the center of a village or along main corridors of travel (i.e., the route along the Kerio River) (**Fig 3A and 3B**, **S1 Fig**). Overlaying the tracks with satellite imagery showed that, while some points were logged along the Kerio River or the shore of Lake Turkana, many of the trajectories moved away from these larger sources of water.

At the individual traveler level, four travel patterns emerged from the GPS data (**Fig 4**, **S2–S5 Figs**). The most common trip type was Long Term with 41.4% (24/58) of travelers, followed by Transient (34.4%, 20/58), Static (19.0%, 11/58), and Day trips (5.2%, 3/58) (**Table 1**). As expected, travelers with Static trips had the fewest unique campsites logged (1, IQR 1–2) and covered the shortest distance (a median total distance of 33.5 kms, IQR 23.0–54.9). With a median duration of 70 days, the Static trip durations were generally longer than the other three trips (~55 days). At the other end of the spectrum, travelers with Transient trips logged the most unique campsites (17, IQR 11–32.8) and traveled the furthest (278.5 km, IQR 186.3–557.4). While travelers conducting Long Term trips logged more unique night locations than Day trips (10.5 vs 1), they logged fewer kilometers on average than Day trips (106 vs 157km). Of the four female travelers carrying GPS loggers, three were recorded conducting Long Term trips and one a Static trip (**Table 1**).

To further characterize these trip patterns, travel history from the surveys was incorporated. For most trip types, travel surveys tended to underestimate the number of campsites calculated from the GPS loggers (**Table 1**) and did not distinguish between the distances covered (i.e., using travel time to a camp as a proxy for distance). These differences possibly reflect recall bias and suggest that a travel survey alone might not detect the nuances of different trip types. The travel surveys collected from travelers who conducted Static trips indicated that these travelers visited 1 campsite which took 2–3 days to reach, suggesting that these travelers may have left their GPS loggers at their homestead while they traveled with the herd (**Table 1**); however, this cannot be confirmed. The median age was similar for Long Term, Transient, and Static trips (36–38 years), but was lower for Day trips (26 years). The majority of travelers for each trip type reported non-household members near their campsites, with a median of 4–6 people reported as near the campsites for all trip types. The majority of travelers on all trip types

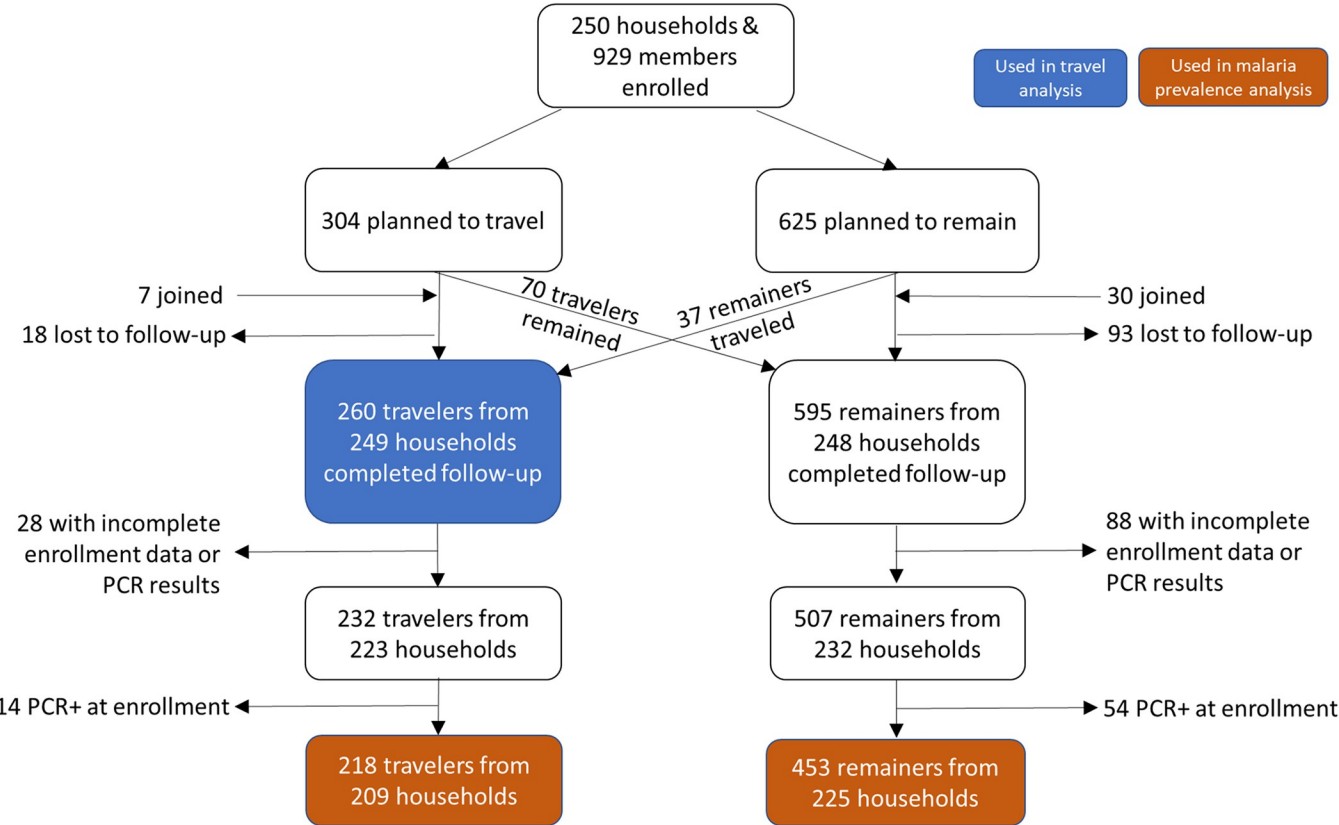

**Fig 2. Diagram of inclusion and exclusion for different analyses.** For the travel analysis, any traveler who provided trip information was included. For the malaria analysis, remainers and travelers had to provide complete information for both enrollment and follow-up as well as test negative for PCR at enrollment. Accounting for the fact that some households had travelers and/or remainers included in the analysis, there was a total of 242 households represented in the malaria prevalence analysis.

reported traveling with goats (≥95) and sheep (≥80%) and having an open water source near their camp (≥ 83.3%).

## Malaria study population

To understand how traveling with the herd might impact infectious disease exposure, we compared the prevalence of malaria in travelers with that of remainers. From the 250 participating households, a total of 929 members were enrolled, consisting of the 304 members who planned to travel and 625 members who planned to remain. At follow-up, 18 travelers and 93 remainers were lost to follow-up, 70 people who planned to travel ended up remaining, 37 people who planned to remain ended up traveling, and 7 new travelers and 30 new remainers joined the study, resulting in 260 traveling members and 595 remaining members surveyed after seasonal travel (**Fig 2**). Our goal was to measure infections acquired during the travel period so we only included those *without* malaria at baseline in the subsequent analysis. We excluded individuals with no baseline data (n = 94), incomplete PCR results at either timepoint (enrollment n = 15, follow-up n = 7), and malaria at baseline by PCR (n = 68). This resulted in 218 travelers and 453 remainers from 242 households in the analysis sample.

From the analyzed cohort, there was a median of 3 members (IQR: 3–5), 1 traveler (IQR 1:1) and 2 remainers (IQR 1:3) enrolled per household (**Table 2**). At least one child ≤15 years old was enrolled for 59.4% (142/239) households. The majority of households reported using open water sources for drinking and cooking (65.7%, 157/239), and relied on livestock as their

**Table 1. Characteristics of travelers and trip-types, overall and stratified by GPS carrier status.** % (number/N); median (IQR).

| | All travelers (N = 260) | Non-GPS carriers (N = 202) | All GPS carriers (N = 58) | GPS trip types | | | |
| --- | --- | --- | --- | --- | --- | --- | --- |
| | | | | Long Term: (N = 24) | Transient: (N = 20) | Daytrips: (N = 3) | Static: (N = 11) |
| **Traveler details** | | | | | | | |
| Male | 87.7 (228/260) | 86.1 (174/202) | 93.1 (54/58) | 87.5 (21/24) | 100 (20/20) | 100 (3/3) | 90.9 (10/11) |
| Age (years) | 31 (19–40) | 30 (19–40) | 37 (30–45) | 37 (30.0–48.5) | 36 (28.8–36.0) | 26 (22–44) | 38 (33–46) |
| Catchment area | | | | | | | |
| Kangirisae | 24.2 (63/260) | 23.8 (48/202) | 25.9 (15/58) | 29.2 (7/24) | 30 (6/20) | 0 (0/3) | 18.2 (2/11) |
| Lowae | 26.2 (68/260) | 26.2 (53/202) | 25.9 (15/58) | 37.5 (9/24) | 5 (1/20) | 33.3 (1/3) | 36.4 (4/11) |
| Nakurio | 25.4 (66/260) | 25.2 (53/202) | 22.4 (13/58) | 8.3 (2/24) | 35 (7/20) | 0 (0/0) | 36.4 (4/11) |
| Kerio | 24.2 (63/260) | 23.8 (48/202) | 25.9 (15/58) | 25 (6/24) | 30 (6/20) | 66.7 (2/3) | 9.1 (1/11) |
| **Reported trip details** | | | | | | | |
| Trip duration (days) | 57.2 (42.2–76.2) | 57.2 (42.2–75.2) | 57 (39.3–90.3) | 56 (41.5–77.5) | 55 (33.8–83.8) | 53 (41–92.5) | 70.0 (47.0–104.5) |
| Camps reported | 1 (1–2) | 1 (1–1) | 1 (1–2) | 1 (1–1.3) | 2 (1–2) | 3 (2–3) | 1 (1–1) |
| Travel time to camp (days) | 2–3 (1:2–3) | 2–3 (1:2–3) | 2–3 (1:2–3) | 2–3 (1:2–3) | 2–3 (1:2–3) | 1–2 (<1:2–3) | 2–3 (2–3:2–3) |
| Non-HH members present | 83.5 (217/260) | 81.7 (165/202) | 89.7 (52/58) | 91.7 (22/24) | 85.0 (17/20) | 66.7 (2/3) | 100 (11/11) |
| People at camp (#) | 4–6 (1–3: 7–10) | 4–6 (1–3: 7–10) | 4–6 (4–6: 7–10) | 4–6 (4–6: 7–10) | 4–6 (4–6: 7–10) | 4–6 (1–3: 7–10) | 4–6 (1–3: 4–6) |
| Nearby water source* | | | | | | | |
| Open[1] | 86.9 (226/260) | 87.6 (177/202) | 84.5 (49/58) | 83.3 (20/24) | 85.0 (17/20) | 100 (3/3) | 90.9 (10/11) |
| Closed[2] | 30.4 (79/260) | 29.7 (60/202) | 32.8 (19/58) | 42.7 (10/24) | 20 (4/20) | 33.3 (1/3) | 36.4 (4/11) |
| Animals traveled with | | | | | | | |
| Goats | 99.2 (258/260) | 99.5 (201/202) | 98.3 (57/58) | 100 (24/24) | 95 (19/20) | 100 (3/3) | 100 (11/11) |
| Sheep | 88.0 (229/260) | 88.1 (178/202) | 87.9 (51/58) | 87.5 (21/24) | 80 (16/20) | 100 (3/3) | 100 (11/11) |
| Camels | 9.2 (24/260) | 9.4 (19/202) | 8.6 (5/58) | 12.5 (3/24) | 5 (1/20) | 0 (0/3) | 9.1 (1/11) |
| **GPS details** | | | | | | | |
| Campsite changes | – | – | – | 4 (2–8.3) | 17 (11–32.8) | 0 (0–3.5) | 0 (0–1) |
| Campsites logged | – | – | – | 3 (3–5.3) | 10.5 (7.3–18) | 1 (1–3) | 1 (1–2) |
| Total distance between camps (km) | – | – | – | 29.0 (10.6–53.2) | 87.8 (69.5–210.3) | 1.6 (1.2–34.1) | 2.1 (1.5–5.8) |
| Total distance traveled (km) | – | – | – | 106.8 (35.9–156.5) | 278.5 (186.3–557.4) | 157.4 (130.5–186.1) | 33.5 (23.0–54.9) |

* Water source types were not mutually exclusive–both could be reported by participant

[1]Open water source: River, lake, dam, spring, hand dug water pit

[2]Closed water source: Tap water, well or borehole

primary source of income (63.2%, 151/239). All households reported ownership of livestock, with most owning goats (99.6%, 237/238) and sheep (93.7%, 223/238). All households reported they had no available bednets.

Remainers tended to be female (67.3%, 305/453) with a median age of 19 years (IQR: 10–32), relative to travelers who were predominantly male (86.2%, 188/218) with a median age of 30.5 years (IQR: 21.3–42.0) (**Table 3**). Remainers included a larger proportion of children ≤15 years (Remainers: 41.3%, 187/453, Travelers: 14.2%, 31/218), while travelers had a larger proportion of adults > 40 years (Remainers: 11.5%, 52/453, Travelers: 26.6%, 58/218). Participants were asked to report any symptoms experienced on the day of follow-up as well as any illnesses that occurred between enrollment and follow-up to gain insight on their health throughout the travel period. Most remainers (96.5%, 437/453) and travelers (93.1%, 203/218) did not report any symptoms at follow-up and few reported being sick during the travel period, although

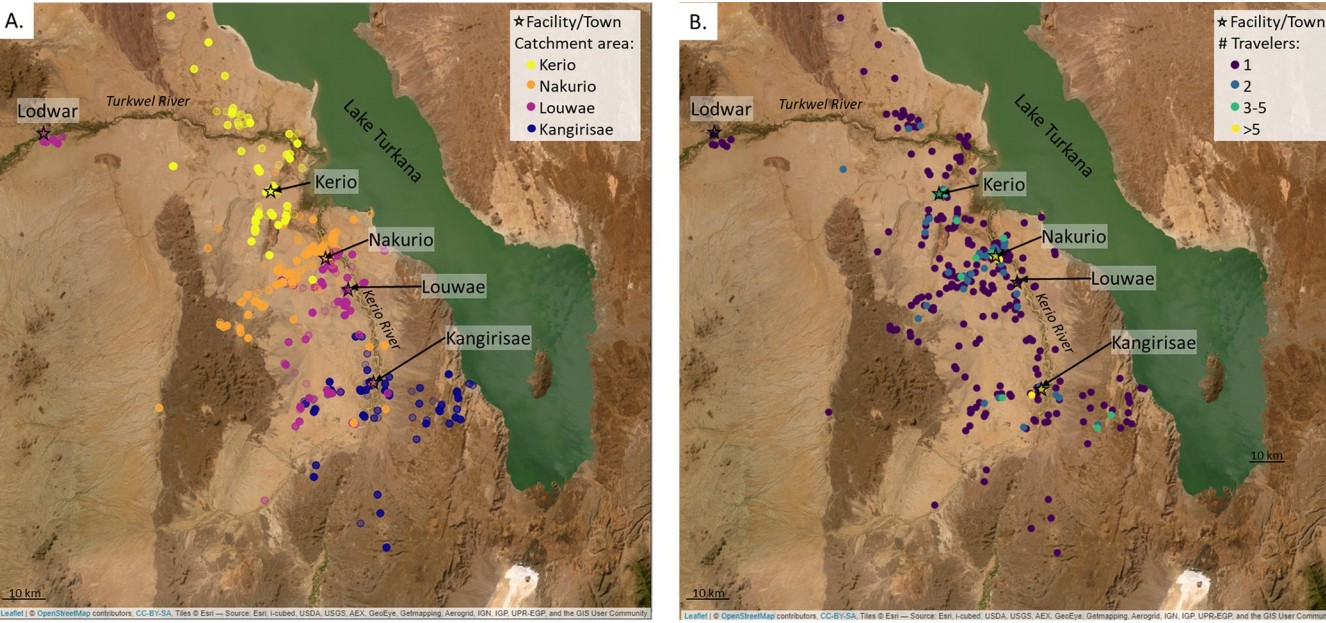

**Fig 3. Population level trip characteristics.** (A) Night locations (presumably campsites), stratified by traveler's catchment area shows regionality in locations visited. (B) Night locations, colored by the number of households logged at a given location to show areas commonly visited. Satellite image from Leaflet package in R, sourced by Esri. See **S1 Fig** for maps with all points (day and campsites) logged.

travelers were twice as likely to report being sick since enrollment (Travelers: 3.7%, 8/218, Remainers: 1.8%, 8/452). Of those who reported being sick since enrollment, 75% (6/8) of travelers self-medicated (i.e., took medicine or herbs from home or bought medicine from a pharmacy) while 75% (6/8) of remainers tended to visit a health facility. Of the participants who tested for malaria when feeling sick, travelers had a higher reported malaria positivity rate (Travelers: 100%, 3/3, Remainers: 50%, 1/2), although the number of observations was very small. Travelers who reported feeling recently sick reported taking medicine less often than remainers (Travelers: 50%, 4/8, Remainers: 75%, 6/8).

## Malaria analysis

After travelers returned, 9.2% (20/218) of travelers (9.3%, n = 16/172 of non-GPS carriers and 8.7%, n = 4/46 of GPS carriers) and 4.4% (20/453) of remainers tested positive for malaria by PCR (**Table 3**). The prevalence of new infections was higher in travelers than remainers across gender, age group, catchment area, and type of water source (open or closed). While infection was similar for both male and female travelers (9–10%), it was twice as high in male remainers (6.8%, 10/148) than female remainers (3.3%, 10/305). Children ≤ 15 years had similar malaria burdens, regardless of their travel status (Travelers: 6.5%, 2/31; Remainers: 6.4% 12/187); however, new malaria infections increased with age for travelers (up to 12.1% in >40-year-olds) and decreased with age for remainers (down to 1.9% in >40-year-olds). The catchment area with the lowest number of infections was Louwae for both groups (7.1% (4/56) of travelers and 3.1% (4/127) of remainers) while the catchment areas with the highest was Nakurio for travelers (11.1% (6/54)) and Kerio for remainers (5.8% (7/120)). There was one month where malaria was more common in remainers than in returning travelers; otherwise, malaria infection in travelers was similar to or greater than the prevalence in remainers in all the study months (**S6 Fig**). The prevalence of new infections reached a maximum of 15.5% (13/84) in travelers returning in July and 8.3% (2/24) in remainers after trips concluding in May.

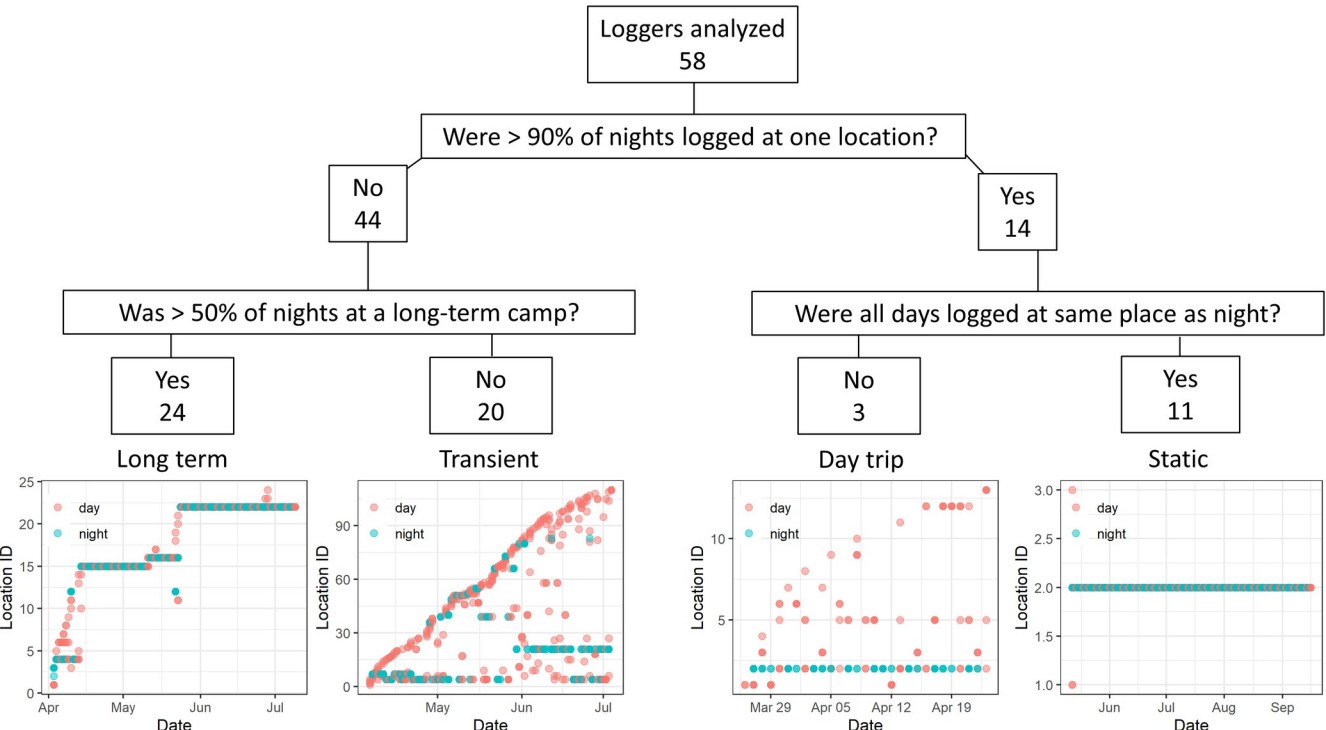

**Fig 4. Individual trip patterns from GPS logger data.** Using GPS logger data, we defined trajectories based on night (blue) and day (red) locations logged by each traveler. Both Long Term and Transient travelers logged a variety of night and day locations; however, Long Term travelers tended to spend $\geq 7$ consecutive nights at each campsite whereas Transient travelers tended to spend $< 7$. Day trip and Static travelers both spent $> 90\%$ of their nights at the same location. They differed by the way Day trip travelers visited different locations during the day, while Static travelers logged all night and day points at the same location. The bottom four plots are tracks from four individuals that exemplify the different types of trip patterns. Tracks from all travelers are found in **S2–S5 Figs**.

Individual-level models identified characteristics associated with acquiring an infection over the course of traveling with the herd (**Table 4**). Univariate analysis revealed that the odds of getting malaria was more than two times greater in travelers (OR = 2.19, 95% CI: 1.15–4.18) relative to remainers and in males (OR = 2.16, 95% CI: 1.11–4.40) relative to females. Age group, catchment area, and water source did not appear to be statistically significant factors. A multivariate model was built to adjust for three factors: travel status (the variable of interest), age (a known risk factor for malaria [40]), and gender, which was a significant factor in univariate analysis. The adjusted model indicated that travel status and gender were no longer statistically significant; however, this can be attributed to the relatively small sample size and the low number of infections which limited the power of the study. The general trend suggests gender and travel are still associated with an increased odds of malaria (aOR = 1.91 (0.86–4.32) for travelers relative to remainers and aOR = 1.54 (0.70–3.48) for males relative to females). While trip type could not be characterized for all travelers and thus was too sparse a factor to include in the model, it is interesting to note that new malaria infection in GPS carriers was 0% after Static (0/10) or Day trips (0/2), 5.6% (1/18) after a Transient trip, and 13.6% (3/22) after a Long Term trip.

## Discussion

Understanding the relationship between mobile communities' travel patterns, health-care seeking behaviors, and risk for disease transmission is critical for both informing intervention

**Table 2. Household characteristics of members included in the malaria analysis.** % (number/N); median (IQR).

| | Household (N = 239) |
|---|---|
| People per household enrolled | 3 (3–5) |
| Travelers per household enrolled | 1 (1–1) |
| Remainers per household enrolled | 2 (1–3) |
| At least 1 child $\leq$ 15 years enrolled | 59.4 (142/239) |
| Number of nets per household | 0 (0–0) |
| **Catchment area** | |
| Kangirisae | 25.5 (61/239) |
| Kerio | 25.5 (61/239) |
| Louwae | 23 (55/239) |
| Nakurio | 25.9 (62/239) |
| **Nearby Water source*** | |
| Open | 65.7 (157/239) |
| Closed | 45.2 (108/239) |
| **Primary income source** | |
| Livestock | 63.2 (151/239) |
| Burning Charcoal | 33.1 (79/239) |
| Weaving | 17.2 (41/239) |
| Farming | 1.7 (4/239) |
| Informally Employed[1] | 2.1 (5/239) |
| **Households own certain livestock** | |
| Goats | 99.6 (237/238) |
| Sheep | 93.7 (223/238) |
| Camels | 26.8 (64/239) |
| Cattle | 8.0 (19/239) |

[1]Informally Employed: Income from a relative or working at a small business

*See Table 1 footnotes for water source details

strategies suitable for their lifestyles and aiding in broader disease elimination campaigns. Here, we quantified travel patterns in four semi-nomadic communities and determined that the prevalence of new infections was twice as high in individuals who traveled with their herd than household members who remained behind. While malaria was acquired at similar proportions for male and female travelers, it was twice as high in males than female remainers, suggesting that gender norms may play a role in exposures around the homestead. Travel patterns in these areas had not been well characterized and we describe local movement of herders that is quite heterogeneous within a small geographic area. Although region of travel is more similar amongst individuals within a catchment than between, the distance, duration and overnight destinations were highly variable across individuals. None of the households reported access to a bednet and few travelers sought treatment from a health facility when they felt sick, consistent with the tendencies for mobile communities to have reduced access to prevention and health care [12,41]. Community-based strategies, where individuals are taught how to detect and manage infections in their communities when access to healthcare is limited, have been successful in other nomadic communities and could help reduce malaria transmission in these communities [12]; however, many of this study's participants who tested positive for malaria were asymptomatic which would make it challenging to know when to take action. Additionally, mobile clinics placed along well traveled routes have been used to provide health

**Table 3. Characteristics of travelers and remainers (all and those who tested PCR+ for malaria after migration) % (number/N); median (IQR).**

| | Traveler | | Remainer | |
|---|---|---|---|---|
| | **All (N = 218)** | **PCR+ (N = # in category)** | **All (N = 453)** | **PCR+ (N = # in category)** |
| **Gender** | | | | |
| Female | 13.8 (30/218) | 10.0 (3/30) | 67.3 (305/453) | 3.3 (10/305) |
| Male | 86.2 (188/218) | 9.0 (17/188) | 32.7 (148/453) | 6.8 (10/148) |
| **Age (yrs)** | | | | |
| ≤15 | 14.2 (31/218) | 6.5 (2/31) | 41.3 (187/453) | 6.4 (12/187) |
| 16–40 | 59.2 (129/218) | 8.5 (11/129) | 47.2 (214/453) | 3.3 (7/214) |
| >40 | 26.6 (58/218) | 12.1 (7/58) | 11.5 (52/453) | 1.9 (1/52) |
| Median age | 30.5 (21.3–42.0) | 31.5 (20.5–46.0) | 19.0 (10–32) | 12.0 (6–26) |
| **Catchment area** | | | | |
| Kangirisae | 23.9 (52/218) | 9.6 (5/52) | 19.4 (88/453) | 4.5 (4/88) |
| Kerio | 25.7 (56/218) | 8.9 (5/56) | 26.5 (120/453) | 5.8 (7/120) |
| Louwae | 25.7 (56/218) | 7.1 (4/56) | 28.0 (127/453) | 3.1 (4/127) |
| Nakurio | 24.8 (54/218) | 11.1 (6/54) | 26.0 (118/453) | 4.2 (5/118) |
| **Water source since enrollment*** | | | | |
| Open | 87.2 (190/218) | 8.4 (16/190) | 65.1 (295/453) | 5.1 (15/295) |
| Closed | 30.7 (67/218) | 11.9 (8/67) | 48.6 (220/453) | 3.2 (7/220) |
| **Medical History** | | | | |
| No symptoms on day screened | 93.1 (203/218) | 7.9 (16/203) | 96.5 (437/453) | 4.1 (18/437) |
| Sick since enrollment | 3.7 (8/218) | 50.0 (4/8) | 1.8 (8/452) | 12.5 (1/8) |
| **Action taken when sick** | | | | |
| Visited health facility | 25.0 (2/8) | 50.0 (1/2) | 75.0 (6/8) | 16.7 (1/6) |
| Self-medicated | 75.0 (6/8) | 50.0 (3/6) | 12.5 (1/8) | 0.0 (0/1) |
| Took malaria test | 37.5 (3/8) | 100.0 (3/3) | 25.0 (2/8) | 50.0 (1/2) |
| Positive malaria test | 100.0 (3/3) | 100.0 (3/3) | 50.0 (1/2) | 100.0 (1/1) |
| Took medicine | 50.0 (4/8) | 75.0 (3/4) | 75.0 (6/8) | 16.7 (1/6) |
| Antimalarial | 37.5 (3/8) | 100.0 (3/3) | 12.5 (1/8) | 100.0 (1/1) |
| Antibiotic | 12.5 (1/8) | 0.0 (0/1) | 62.5 (5/8) | 0.0 (0/5) |
| Pain Killer | 50.0 (4/8) | 75.0 (3/4) | 75.0 (6/8) | 16.7 (1/6) |

*See Table 1 footnotes for water source details

care to patients who would not have had access otherwise [42]; however, this study did not reveal any common routes in these communities which would make it difficult to determine where to place a mobile clinic. Screening travelers upon their return would be a proactive approach, but likely unrealistic, given how often travel occurs, the unpredictable nature of when and where the trips will take place, and the resource constraints on intensifying control efforts. Instead, focusing on preventative measures, such as bednet distribution in more remote areas would be conducive for reducing malaria transmission in travelers who spend most of their nights at the same place (i.e., Day trip or Static trips) and also prevent onward transmission from infected travelers upon their return. For travelers who spend more nights away from their settlements (i.e., Long term or Transient travelers), there is a need for solutions that would easily integrate into their lifestyle, such as insecticide treated clothing [43]. Bednets designed specifically for use among outdoor sleeping populations could also be considered for both remainers and travelers [44]. To improve intervention efficacy and uptake, community health workers should be empowered to regularly raise awareness around why

**Table 4. Factors associated with PCR+ malaria cases.** Significance levels at or below 0.05 indicated by *.

|  | OR (95% CI) | aOR (95% CI) |
|---|---:|---:|
| **Travel Status** |  |  |
| Remainer | ref | ref |
| Traveler | 2.19 (1.15–4.18)* | 1.91 (0.86–4.32) |
| **Age (years)** |  |  |
| ≤15 | ref | ref |
| 16–40 | 0.81 (0.39–1.69) | 0.69 (0.32–1.52) |
| >40 | 1.14 (0.44–2.76) | 0.86 (0.31–2.22) |
| **Gender** |  |  |
| Female | ref | ref |
| Male | 2.16 (1.1–4.40)* | 1.54 (0.70–3.48) |
| **Catchment** |  |  |
| Kangirisae | ref |  |
| Kerio | 1.07 (0.43–2.69) |  |
| Louwae | 0.67 (0.24–1.79) |  |
| Nakurio | 0.99 (0.40–2.54) |  |
| **Water** |  |  |
| Closed only | ref |  |
| Open only | 1.77 (0.87–3.91) |  |
| Open and closed | 0.73 (0.16–2.46) |  |

and how prevention practices should be implemented [45]. Ultimately, maintaining an open dialogue with mobile communities about their needs and co-developing practical solutions is critical to ensure interventions are most useful.

This is in line with other studies indicating that there is no standard way to address malaria transmission in mobile populations due to the wide range of travel patterns and lifestyles exhibited by mobile populations. For instance, studies in Cambodia and Laos have also shown how different mobility patterns can be linked with different risks of malaria in mobile populations, whether they are traveling as forest-goers, migrant workers, or indigenous people traveling to their farms [26,46]. As the world strives for malaria elimination, taking this initial step of understanding the motivation for travel, mobility patterns, and malaria risks of mobile communities that are hard to reach and may still be at risk for transmission is key before proposing and implementing an intervention.

While this study has established that travel patterns can be very heterogeneous both in duration and destination within communities and that malaria is often acquired while traveling with the herd, its limitations have highlighted a number of important research areas that remain to be pursued. First, we still do not fully understand where the travelers were acquiring malaria on their travels. GPS data did not reveal potential transmission hotspots [39] and the satellite imagery accessed in the analysis did not have the spatial nor temporal resolution to pick up on transient water sources travelers may have stayed near. While we assumed that travelers were likely exposed to environments with higher mosquito activity and malaria transmission due to seeking out water for their herds, work needs to be done to characterize the conditions and malaria endemicity of the different locations visited and compare them with the homestead conditions. Part of better characterizing the conditions of the different locations should involve documenting the malaria vector(s) in the region. As a result of malaria being previously assumed to not be endemic in Turkana, malaria vectors in this county have not been well studied. In 2010, Okara *et al.* surveyed the distribution of malaria vectors in Kenya

[47]. Only two surveillance sites were in Turkana county's 100,000 km$^2$ (neither of which were close to this study area) and they were only surveying for *Anopheles arabiensis*, the vector associated with the malaria outbreaks in the Kakuma refugee camp [47,48]. Work is being done to start filling the knowledge gap around malaria vectors in Turkana. For instance, a recent study by Ochomo *et al.* indicates that *Anopheles stephensi* are present in Lodwar, the capital [49]. However, as *A. stephensi* are predominantly found in urban settings, we cannot be certain they were also present in the more rural areas where the participants were located. Given the nascent stage of knowledge around vectors in this region, this study was focused more on determining if traveling with herds was a risk factor for acquiring malaria, regardless of vector. To recommend suitable interventions, an important next step would be to better characterize the vectors and their habitats across Turkana and determine which ones are most likely encountered by semi-nomadic communities.

Additional research is also needed to generalize the relationship between mobility and malaria in Turkana and beyond. This study analyzed ~6% (232/3800) households in the study area over ~7months, thus our sample may be biased towards those who travel earlier in the season or are better known by the CHW. It is unlikely that this bias is related to the outcome but it may affect generalizability. To identify seasonal patterns in mobility and malaria in Central Turkana, a larger cohort would need to be enrolled and followed for at least two years. Moreover, our findings are likely specific to Central Turkana, where movement is less extensive than among nomadic groups in Western Turkana, where households often travel hundreds of kilometers with their entire household and cross into Uganda or Ethiopia (personal communication with local health authorities). Additional studies are needed to characterize these movements, transmission dynamics, and health seeking behaviors before generalizable conclusions can be drawn and opportunities for intervention can be determined. However, based on our findings, we suspect that those who travel with their herds will still have higher risks of malaria exposure.

The limitations of this study include challenges with collecting complete information at enrollment and follow-up, which ultimately lead to a smaller sample size. In addition, it was difficult to know whether the GPS loggers were actually being carried. For instance, the fact that 11 loggers recorded points for multiple weeks within the same 0.5km$^2$ area suggests that they might have been left behind. While this may have led to mis-categorization of some trips, we were still able to categorize three different trip patterns that would be informative for different intervention approaches. Most of the variables we explored were self-reported, therefore we cannot rule out mis-reporting things like nets, livestock, and symptoms, or recall bias for trip details and recent illnesses. As we only screened for malaria at enrollment and follow-up, there is a chance that infections acquired during the migration could have cleared before they were detected at follow-up. This could happen if a participant became infected and sought malaria treatment; however, parasite DNA is detectable by PCR 42–48 days after successful treatment and all participants who reported testing positive for malaria and taking an antimalarial during the travel period also tested positive at follow-up [50]. Infections that were acquired during the trip and left untreated could last for years and thus should have been present at follow-up (the longest trip was 188 days), but we cannot rule out the possibility that some infections may have been below the detection limit of PCR (1000 parasites/mL) [51]. Furthermore, we only screened for *P. falciparum;* yet, recent literature suggests that *P. vivax* is also circulating in this region [52]. If this is true, then the prevalence of malaria in this study (and the region in general) is being under-reported.

In conclusion, this study is one of the few that quantitatively characterizes the mobility patterns and its relationship with disease exposure in mobile communities. We determined that traveling with the herd doubled the odds of acquiring a new malaria infection, relative to those

who remained behind. The different travel patterns identified could be used to inform intervention strategies more suitable to a mobile life-style. Further studies are needed to determine how these observations can be generalized to other disease exposures as well as the role of mobile populations on disease transmission with the broader community.

## Supporting information

**S1 Checklist. Inclusivity in global research.**
(DOCX)

**S1 Text. Using reported departure and return dates instead of the enrollment and follow-up dates.**
(DOCX)

**S2 Text. Sensitivity analysis around definition of "long term campsite".**
(DOCX)

**S1 Table. Using reported departure and return dates instead of the enrollment and follow-up dates.**
(DOCX)

**S2 Table. Sensitivity analysis around definition of "long term campsite".**
(DOCX)

**S1 Fig. Population level trip characteristics.** (A) All locations logged, stratified by traveler's catchment area shows regionality. (B) Campsite locations, colored by the number of households logged at a given location to show areas commonly visited. Satellite image from Leaflet package in R, sourced by Esri.
(TIF)

**S2 Fig. GPS carrying travelers whose trips were categorized as static.** >90% of the night spots were spent at the same night location they were enrolled at, but most day points were logged at different locations.
(TIF)

**S3 Fig. GPS carrying travelers whose trips were categorized as day trips.** >90% of the night spots were spent at the same location and most night and day points were logged at the same location.
(TIF)

**S4 Fig. GPS carrying travelers whose trips were categorized as transient.** <90% of the night spots were spent at the same location and > 50% of nights were spent at transient camps (defined as camps with < 7 consecutive nights spent).
(TIF)

**S5 Fig. GPS carrying travelers whose trips were categorized as long term.** <90% of the night spots were spent at the same location and > 50% of nights were spent at long term camps (defined as camps with >7 consecutive nights spent).
(TIF)

**S6 Fig. Proportion of remainers and travelers who tested PCR+ for malaria after the migration period, stratified by the month the travelers returned.**
(TIF)

## Acknowledgments

We thank our field team, especially Dennis Okoth, Rose Adome, Erastus Lomuria, Benson Lorunye, Ekapeton Jackline, Lorinyo Francis Ethuron, Esinyen Mark, Topuye Ekuom David, Akolonyo Peter Ikao, James Lomaala, Anjeline Atabo, Bosco Nawoto, and all the families who gave their valuable time to participate in this study.

## Author Contributions

**Conceptualization:** Hannah R. Meredith, Amy Wesolowski, Wendy Prudhomme O'Meara.

**Formal analysis:** Hannah R. Meredith.

**Funding acquisition:** Hannah R. Meredith.

**Investigation:** Hannah R. Meredith, Dennis Okoth, Linda Maraga, George Ambani, Tabitha Chepkwony, Lucy Abel, Joseph Kipkoech.

**Methodology:** Hannah R. Meredith.

**Project administration:** Hannah R. Meredith, Joseph Kipkoech.

**Supervision:** Gilchrist Lokoel, Daniel Esimit, Samuel Lokemer, James Maragia, Andrew A. Obala.

**Visualization:** Hannah R. Meredith.

**Writing – original draft:** Hannah R. Meredith.

**Writing – review & editing:** Hannah R. Meredith, Amy Wesolowski, Wendy Prudhomme O'Meara.

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
