## [Decision Letter · Decision Letter 0]

3 Jan 2024

PGPH-D-23-02422

Characterizing mobility patterns and malaria risk factors in semi-nomadic populations of Northern Kenya

Dear Dr. Wendy,

Thank you for submitting your manuscript to PLOS Global Public Health. After careful consideration, we feel that it has merit but does not fully meet PLOS Global Public Health’s publication criteria as it currently stands. Therefore, we invite you to submit a revised version of the manuscript that addresses the points raised during the review process.

We look forward to receiving your revised manuscript.

Kind regards,

Abhinav Sinha, M.D.

Academic Editor

Journal Requirements:

Additional Editor Comments (if provided):

Reviewers' comments:

Reviewer's Responses to Questions

**Comments to the Author**

1. Does this manuscript meet PLOS Global Public Health’s publication criteria? Is the manuscript technically sound, and do the data support the conclusions? The manuscript must describe methodologically and ethically rigorous research with conclusions that are appropriately drawn based on the data presented.

Reviewer #1: Partly

Reviewer #2: Yes

2. Has the statistical analysis been performed appropriately and rigorously?

Reviewer #1: No

Reviewer #2: Yes

3. Have the authors made all data underlying the findings in their manuscript fully available (please refer to the Data Availability Statement at the start of the manuscript PDF file)?

Reviewer #1: No

Reviewer #2: Yes

4. Is the manuscript presented in an intelligible fashion and written in standard English?

Reviewer #1: Yes

Reviewer #2: Yes

5. Review Comments to the Author

Reviewer #1: The paper aims to evaluate mobility patterns and their role in determining the risk of malaria exposure in semi-nomadic populations in Central Turkana, Kenya. The theme is intriguing, and further investigation is warranted in this area, reinforcing the justification for developing this study. The structure is well-defined, encompassing introduction, study area, data, study design, results, discussion, and conclusion. However, I have some concerns regarding the content:

ABSTRACT

- The abstract could emphasize the malaria burden in the country, the study region, and within this mobile group, underscoring the significance of the study for this specific population.

STUDY DESIGN

- Consider addressing malaria seasonality in the study location. Why wasn't the survey conducted throughout the entire year? Does the study period capture the peak of yearly malaria transmission?

- Information needed: Provide details on the conditions where these households are located. Are they more protected to malaria vectors and parasites than other areas these populations travel to?

- Information needed: Discuss the vector in this area and main breeding sites. Detail vector habits (night, day, sunset, dusk, all day) and explore how the vector is related to mobile populations. Are entomologic data available? If not how can this challenge be circumvented?

- In connection with the preceding point, let's revisit the hypothesis: “Since nomadic individuals traveling with their herds tend to congregate at water sources, potential mosquito breeding sites, we hypothesized that the Turkana who travel with their herds are more likely to be exposed to malaria than those who remained in the settled communities.” Has this hypothesis been rigorously tested? Considering the utilized data, it appears that the results might not adequately address this hypothesis.

- Explain the process of household selection and justify why these specific catchment areas were chosen. Provide information on the total population in the selected catchment areas and discuss the sample's representativeness concerning location.

- Discuss why community health workers were responsible for participant identification and enrollment. Were they adequately trained? Why not have research scientists conduct these activities to minimize bias/errors?

- Concern: At enrollment, before the travelers left with their herds, and again at follow-up, after the travelers returned with their herds, participants provided a finger-prick blood sample for a dried blood spot (DBS) and answered a questionnaire detailing their travel and medical history”. Depending on the duration of their absence, malaria parasites might become undetectable upon their return. In the event of infection during travel, symptoms or parasitemia could become indiscernible. How was this challenge addressed, and what impact did it have on the results?

RESULTS

- Table 1. As an illustration, consider the comparison between static and transient groups. The former, characterized by no campsite changes, a smaller distance between camps, and lower overall travel distance, may face an increased risk of malaria infection if they remain static in an area where they are more exposed to the malaria vector. In such cases, their likelihood of being infected with malaria parasites would rise. On the contrary, the transient group has the flexibility to move to other areas with a lower likelihood of infection, thereby protecting themselves from getting infected.

To gain a comprehensive understanding, it is essential to determine the endemicity of the areas where these populations move and associate it with their mobility patterns. Additionally, this information should be compared with the endemicity of the locations where the remainers, who have not moved, are situated. Without such considerations, we risk making assumptions about how mobility influences malaria infection rates, potentially overlooking key conditioning factors that may not have been measured.

- Table 2. Have other vector control strategies been implemented, such as indoor residual spraying? Is malaria treatment readily available, and if so, how accessible is it? In terms of water sources, are there locations where both open and closed water sources are nearby, or is it a situation where only one type is available and not the other?

- Table 3. We aim to clearly understand the prevalence of malaria among travelers during both the baseline and follow-up surveys. Additionally, we seek to determine the malaria prevalence among remainers at both baseline and follow-up.

- Table 3. To assess whether there is a statistically significant difference in characteristics between travelers and those who remained, a chi-square test could be employed to compare PCR+ results. This test would help determine if the observed percentages are statistically significant, providing insight into the distinctions between individuals who acquired malaria while traveling versus those who stayed in one location.

- Table 3. Clarify the term "sick since enrollment" in Table 3 and explain the criteria for sickness.

- Table 3. What does "sick since enrollment" entail? Does it imply that individuals retained symptoms from the enrollment survey until the follow-up survey? This interpretation appears puzzling. Perhaps the intended expression is "sick at enrollment." Furthermore, the term "sick" requires clarification – does it refer to being diagnosed with malaria, and is this diagnosis based on PCR conducted by the research project?

- Table 4. Discuss the lack of statistically significant results in the multivariate model. Consider exploring other variables not used in the modeling (trip types, catchment area, water source, income source, etc.).

- Table 4. The analysis excluded trip types due to small group sizes. Ideally, a more effective approach would involve consolidating trip types into two groups, each with a higher population. The isolated inclusion of day trips, with only three observations, may not provide meaningful insights on its own, potentially hindering the identification of patterns. If the number of observations per trip type is insufficient, the study's design could benefit from an enhanced enrollment process. Alternatively, reconsidering the feasibility of conducting statistical modeling analyses might be warranted.

DISCUSSION/CONCLUSIONS

- Expand the limitations section to include lines 341-359.

- Develop policy implications further, in line with the introduction's emphasis on tailoring surveillance and intervention strategies for unique populations.

- Enhance the references, providing more information on the relationship between malaria and population mobility, beyond Africa.

I commend the authors for their efforts in developing this paper, and I hope that these suggestions will assist them in augmenting their contribution to the scientific literature.

Reviewer #2: The MS is well written the GPS technology was used to track the population. Authors have tried to show that infection is acquired by the travelers. some of the observations on the MS are as follows:

Using the GPS was a problem to the human to carry them however, some of the GPS should have been fixed on livestock that might be helpful in case they forgot to carry with them.

In the Methods it is not clear that RDTs were used or not if not : Why RDTs were not used to check any parasitiemea to treat any positive case during collection of Filterpaper blot this would have reduced the transmission in case of positivity

lines 250-252 how was malaria tested in the individuals through PCR only "?. how can it be sure they have not acquire infection from start point have if they were not sampled 10 days after also?

lines 316-318 why there was higher malaria in males remainers while gender positivity insignificant in traveler

lines 357-358 : it was mentioned that hotspot were identified using GPS how ? and secondly malaria is transmitted by infected mosquitoes whether and how infected mosquitoes were available to these travelers are the areas where they stay were infection was already occurring?

overall the study could show the trac of nomadic population and need for their screening whereas was not able to show the areas where probably infection was possibly coming.

the study excluded PV which is even though with complication of relapse it is obvious and make sense to exclude Pv.

there was a need to assess the malaria situation of the areas where these nomadic population was staying during their movements.

Paper needs cognizing it seems lengthy paper.

6. PLOS authors have the option to publish the peer review history of their article (what does this mean?). If published, this will include your full peer review and any attached files.

**Do you want your identity to be public for this peer review?** For information about this choice, including consent withdrawal, please see our Privacy Policy.

Reviewer #1: No

Reviewer #2: **Yes: **Himmat Singh

---

## [Editor Report · Decision Letter 1]

14 Feb 2024

Characterizing mobility patterns and malaria risk factors in semi-nomadic populations of Northern Kenya

PGPH-D-23-02422R1

Dear Dr Wendy,

We are pleased to inform you that your manuscript 'Characterizing mobility patterns and malaria risk factors in semi-nomadic populations of Northern Kenya' has been provisionally accepted for publication in PLOS Global Public Health.

Best regards,

Abhinav Sinha, M.D.

Academic Editor